# Biomimetic caged platinum catalyst for hydrosilylation reaction with high site selectivity

Ganghuo Pan[1], Chunhua Hu [2], Song Hong [3], Huaping Li[1], Dongdong Yu[1], Chengqian Cui [1], Qiaosheng Li[1], Nianjie Liang[1], Ying Jiang [1], Lirong Zheng[4], Lei Jiang[1] & Yuzhou Liu [1,5✉]

Natural enzymes exhibit unparalleled selectivity due to the microenvironment around the active sites, but how to design artificial catalysts to achieve similar performance is a formidable challenge for the catalysis community. Herein, we report that a less selective platinum catalyst becomes highly active and selective for industrially relevant hydrosilylation of a broad range of substrates when a porous cage ligand is used for confinement around the catalytic active site. The catalyst is more than ten times more active than Karstedt's catalyst while being recyclable. Properties such as size-selective catalysis and Michaelis-Menten kinetics support the proposed enzyme-like model. This biomimetic catalyst exhibits remarkable site-selectivity through the cage's confining effect, which amplifies small steric differences into dramatic reactivity changes for similar functional groups within a molecule.

[1] School of Chemistry, Beihang University, Beijing 100191, PR China. [2] The Department of Chemistry, New York University, 100 Washington Square East, New York, NY 10003-6688, USA. [3] Center for Instrumental Analysis, Beijing University of Chemical Technology, Chaoyang, Beijing 100029, PR China. [4] Beijing Synchrotron Radiation Facility, Institute of High Energy Physics, Chinese Academy of Sciences, Beijing 100049, PR China. [5] Beijing Advanced Innovation Center for Biomedical Engineering, Beihang University, Beijing 100191, PR China. ✉email: liuyuzhou@buaa.edu.cn

Pt-catalyzed hydrosilylation is one of the largest industrial applications for homogeneous catalysis, and could convert simple alkene and silane molecules into high-value silicon-based materials with wide application in adhesives, coating, drug development, and many other areas[1,2]. For example, (n-octyl)Si(OEt)$_3$ alone was manufactured through hydrosilylation on a scale of over 6,000 tonnes per year[3]. Use of multiple-functional alkenes as the substrates can lead to hydrosilylation product with one or more functional groups left for broader application space, but this has been impeded by the non-selective nature of the commonly used Karstedt's catalyst, which usually leads to unisolable mixtures or undesired products. Several catalysts can improve the site-selectivity for special cases, but suffers from problems associated with the demanding reaction conditions, moderate selectivity, and limited substrates[4–7]. Realizing high site-selectivity for hydrosilylation with a broad scope of substrates and high activity has been one unmet goal[8]. Inspired by natural metalloenzymes with trapped transition metal atoms inside peptide assemblies, many chemists have endeavored to architect caged catalysts[9–14] from accelerating the reaction rate[15–17], improving selectivity[18–20] to alternating reaction pathway[21,22], but few of such catalysts has been known to possess both high selectivity and high activity while tolerating a wide range of substrates. Here we would like to report our work of trapping Pt catalysts inside an artificial cage assembly, and this biomimetic approach led to a potent catalyst with high activity and high site selectivity for a wide range of multiple functional substrates, generating many silanes hardly accessible through any other method. Beyond the immediate utility of this catalyst to benefit synthesis of fine chemicals and the siloxane industry, our strategy of synthesizing it could also provide inspiration for designing local environment around other catalytic center for high selectivity.

## Results

**Synthesis and characterization of COP1-T-Pt.** We created a vinyl containing organic cage compound (**COP1-T**, Fig. 1) based on a truncated octahedron model, which was then used to trap Pt atoms through the alkene groups. Based on a previously reported pyridine-based tied cage compound[23], we proposed a tied cage with free carboxylic acid groups since we anticipated the hydrogen bonding between carboxylic acid groups could prevent the cage from collapsing. The disk-like C3-symmetric ligand **L1** comprised a central hexaphenylbenzene which was alternately para-substituted with 3-benzoicacid and dimethyl(7-octen-1-yl) silyl groups as shown in Fig. 1. Reacting **L1** with Cu(NO$_3$)$_2$ in DMF (Supplementary 1-1 to 1–5 and Supplementary Figs. 6–11 for experimental details including full characterization for **L1**) led to the formation of truncated octahedron cages with a diameter of around 3.5 nm through the Cu-carboxylate paddle-wheel connection (**MOP1**)[24]. The structural identity and purity of **MOP1** were confirmed by single-crystal X-ray diffraction (Supplementary Table1, Fig. 12, Movie 1 and supported cif file), MALDI-TOF (Supplementary Fig. 13), element analysis (EA), and high resolution tunneling electronic microscopy (HR-TEM) imaging (Supplementary Fig. 14).

The six copper paddle-wheel connectors and eight **L1** ligands occupied the six vertexes and eight faces of the octahedron, respectively, and as a result, two dimethyl(7-octen-1-yl)silyl groups from adjacent **L1** lied close to each other, and this would allow metathesis reaction to happen between them. **MOP1** was then tied up through metathesis reaction between the dangling terminal alkenes by Grubbs II catalyst in THF to give **MOP1-T**. Successful tying was confirmed by IR (Supplementary Fig. 15), [1]H-NMR (Supplementary Fig. 16) and MALDI-TOF analysis (Supplementary Fig. 17) with the loss of 24 terminal CH$_2$ groups

per cage. After copper removal by Na$_2$H$_2$EDTA, **MOP1-T** was transformed to **COP1-T** as white powder (1.05 g, overall 85% yield based on **L1**, [1]H/[13]C-NMR in Supplementary Figs. 18,19, MALDI-TOF in Supplementary Fig. 20). Interestingly, gas sorption experiments showed that **COP1-T** unusually maintained its porosity after solvent removal, with a measured pore size of 1.8 nm and BET surface area of around 1000 m$^2$/g, highlighting the stabilization effect of the carboxylic acid groups. (Supplementary Fig. 21, estimated values of 1.6 nm and 1300 m$^2$/g for **MOP1**, Supplementary ref. [7])

**COP1-T** was mixed with 4 equivalent Na$_2$PtCl$_4$ salt in THF, and subsequently reduced with 100 equivalents of dimethylphenylsilane to produce **COP1-T-Pt**. MALDI-TOF analysis showed the presence of four platinum atoms per cage (Supplementary Fig. 22). [1]H-NMR spectroscopy indicated the interaction between the alkene groups and platinum centers (Supplementary Fig. 23). Dynamic light scattering (DLS) and diffusion ordered spectroscopy ([1]H-DOSY) showed **COP1-T-Pt** had a similar size as **COP1-T** (Supplementary Figs. 24–26), indicating the absence of cage aggregation, which would be unavoidable if the platinum atoms stayed outside of **COP1-T**. Aberration-corrected high-angle annular dark-field scanning TEM (ACHAADF-STEM) of **COP1-T-Pt** located the discrete Pt tetramer spanning a length of around 2.74 nm, smaller than the 3.5 nm cage size (Fig. 1, Supplementary movie 2 and Fig. 28). These characterization results reflected the unique single-atom distribution of four Pt atoms within the cage of **COP1-T**.

**High catalytic efficiency of COP1-T-Pt.** **COP1-T-Pt** exhibited high catalytic efficiency in the hydrosilylation reaction between triethoxysilane and 1-hexene with a record-high TOF of 78,000 h$^{-1}$, over performing common Karstedt's catalyst (6,400 h$^{-1}$) by around 12 times, with an activation energy of around 40 kJ/mol (Supplementary Fig. 29). In addition, our catalyst can be recycled for at least five times with less than 10% loss of the catalytic efficiency (Supplementary Figs. 30, 31), and the DLS and [1]H-DOSY spectrum for the used catalyst remained unchanged, indicating the absence of particle aggregation which is however commonly observed for Karstedt's catalyst after only a single use. ACHAADF-STEM imaging (Supplementary Fig. 32) also showed the atomic isolation of Pt atoms in **COP1-T-Pt** after five uses.

[1]H-DOSY analysis revealed the ability of **COP1-T** to form adducts with alkene and silane molecules (Supplementary Fig. 33), an indication that the cage trapped rehighactants prior to reaction, as enzymes do in nature. As shown in Fig. 2b, kinetic study of the hydrosilylation between triethoxysilane and 1-hexene to produce n-hexyltriethoxysilane (compound **1**) catalyzed by **COP1-T-Pt** (Supplementary Fig. 34) revealed the Michaelis-Menten behavior which has never been reported for hydrosilylation reaction catalyzed by Pt catalysts[1,25–27]. Additional supporting evidence was obtained from the blocking experiment. The carboxylic acid groups resided at the opening of the cage, and placement of bulky ions around them should have adverse effect on the reaction rate by blocking the substrates from entering the cage. In fact, neutralization of the acid groups with N, N, N-trimethyl-1-adamantammonium hydroxide, which had a large size, almost completely inhibited the reaction (Supplementary Fig. 35). Such inhibiting phenomenon wouldn't be observed if the platinum atoms were outside of the cage.

**XAS characterization of COP1-T-P prior to, during, and after reaction.** The local electronic structure and coordination geometry of **COP1-T-Pt** were investigated via synchrotron X-ray absorption spectroscopy (XAS) on Pt L3-edge in the solution of

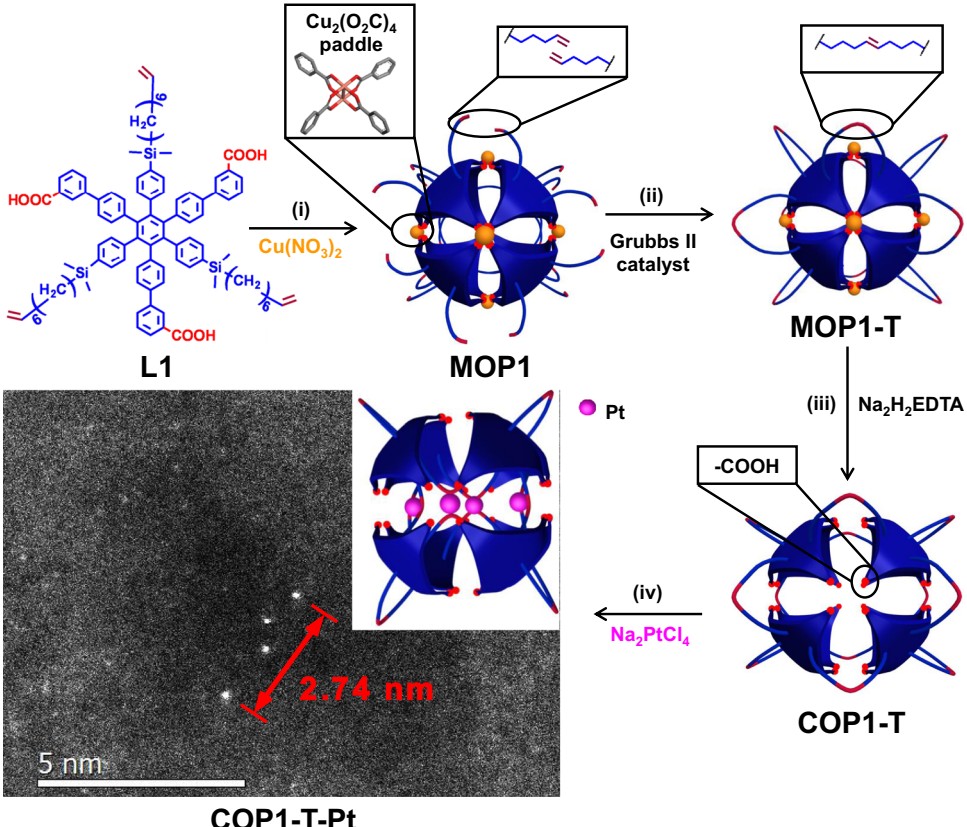

**Fig. 1 Design and Synthesis of COP1-T-Pt.** Synthesis of **COP1-T-Pt** (i) Cu(NO₃)₂, DMF, 60 °C, 24 h, 95% yield (ii) 1 mol% Grubbs II catalyst, THF, 90% yield (iii) Na₂H₂EDTA, THF, H₂O, 99% yield (iv) 4 eq. Na₂PtCl₄ in THF, followed with reduction by 100 eq. dimethylphenylsilane. For **COP1-T-Pt**, among various characterization, the ACHAADF-STEM image was shown with the schematic representation as the insert.

xylene/THF (9:1) prior to reaction. XAS was also performed on the reaction mixture frozen at around 50% substrate conversion (estimated by GC) for in-situ information. Additionally, the final reaction mixture was also analyzed. Three samples were designated as **COP1-T-Pt** (prior to, during, and after).

Prior to reaction, platinum exhibited the oxidation state between 0 and +2 based on the white line analysis (see Fig. 2c) and XPS spectrum (Supplementary Fig. 36)[28], and this indicated that the Pt atom was partially reduced by the excessive dimethylphenylsilane prior to reaction. $k^2$ weighted Fourier-transformed extended X-ray absorption fine structure (FT-EXAFS) revealed the presence of Pt-C (1.74 Å)[27] and Pt-Cl (1.90 Å)[29] bonds (Fig. 2d), which was due to the coordination of alkene groups to the Pt atoms. The absence of Pt-Pt bonds was consistent with single-atom Pt distribution shown by the ACHAADF-STEM imaging. Quantitative EXAFS fitting curves at R space (Fig. 2e) revealed the presence of two coordinating alkene groups and a Cl atom per Pt atom, and this model was validated by comparison with other proposed models in terms of the fitting factors (Supplementary Fig. 37). Further validation and additional information were obtained by the *FDMNES* simulation (Fig. 2h and Supplementary Fig. 38), which also indicated the presence of weakly coordinating silane groups, in addition to the alkene groups, to the Pt atom, and this is not unexpectedly given the excess of silanes used to reduce the Pt atoms prior to reaction.

XAS analysis for the frozen reaction mixture revealed the presence of Pt-C (1.60 Å) and Pt-Si (2.07 Å)[27] bonds simultaneously with an oxidation state higher than **COP1-T-Pt** (prior) (likely to be +2). Both quantitative EXAFS fitting (Supplementary Fig. 39) and *FDMNES* simulation (Fig. 2i and Supplementary Fig. 38) supported the structure as shown in Fig. 2f, which was

actually the key intermediate for the Chalk-Harrod mechanism as shown in Supplementary Fig. 40. This represented an important XAS validation of this important intermediate proposed for Chalk-Harrod mechanism. This intermediate would prevent the formation of the side product of vinyl silane[30], and was consistent with our observation of negligible formation of vinyl silane in our reaction (around 10% for Karstedt's catalyst).

After the reaction, platinum existed as zero-valence with the presence of Pt-C (1.69 Å) bonds only, and quantitative EXAFS fitting and *FDMNES* simulation supported the model of single-atom platinum stabilized with three alkene groups (Fig. 2g, j and Supplementary Figs. 39, 41), quite similar to that for Karstedt's catalyst. The broad shape indicated that the platinum atoms might exist dynamically, for example, shuttling between different alkene-coordinating sites, due to the existence of multiple alkene groups each cage. The stabilization effect by these multiple alkene groups echoed with the recycling use and maintained high activity for **COP1-T-Pt** during catalysis discussed above.

**Size-selective catalysis of COP1-T-Pt.** Due to the confined environment, **COP1-T-Pt** exhibited remarkable size-selective catalysis as shown in Fig. 3 by varying the size of both substrates. For dimethylphenylsilane, as the sizes of alkenes increase from 1-hexene, allylbenzene, styrene to 4-tert-butyl-styrene, the yields of products (compound **2–5**, Supplementary Figs. 43–46) after 24 h gradually decreased from 97%, 95%, 92% to 60%. When bulky 9-vinylanthracene was used, no hydrosilylation product (compound **6**) was observed at all. Large triphenylsilane failed to generate any additional product even with 1-hexene.

And this was actually confirmed by dynamic simulation, which showed that large triphenylsilane encountered significantly larger force when entering the cage than dimethylphensilane, supporting

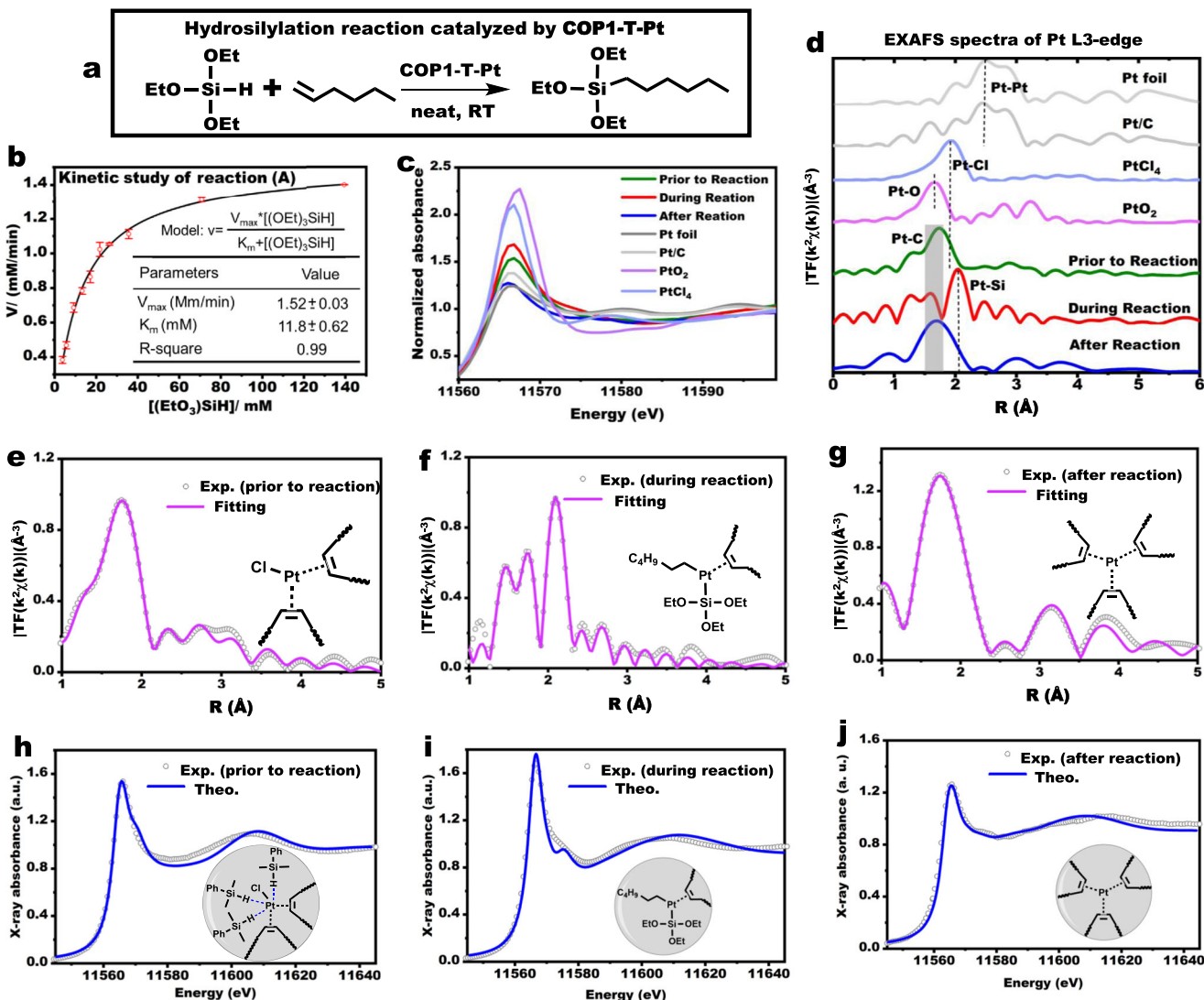

**Fig. 2 In situ XAS characterized of COP1-T-Pt catalysis Hydrosilylation. a** Hydrosilylation reaction between triethoxysilane and 1-hexene catalyzed by **COP1-T-Pt**. **b** Kinetic study in neat 1-hexene at room temperature, and the concentration of **COP1-T-Pt** was $1.67 \times 10^{-6}$ M. The initial reaction rate was measured when the conversion of 1-hexene was less than 2%. The error bar represented the standard error for each initial reaction rate calculated with ten points (Supplementary Fig. 34). **c** Normalized Pt L3-edge XANES spectra of different samples. **d** Fourier transformed (FT) $k^2$-weighted $\chi(k)$-function of the EXAFS spectra for Pt L3-edge spectra. **e–g** Quantitative EXAFS fitting curves at R space for **COP1-T-Pt** prior to reaction, during reaction, and after the reaction with the models shown in each panel. **h–j** Comparison between the experimental Pt L3-edge XANES spectra of **COP1-T-Pt** (prior to, during, after reaction) and the theoretical spectra calculated for the corresponding depicted structures. Since XAS only probed the immediate environment around the Pt centers, 1-propene was used instead for succinctness.

the size-selectivity (Supplementary Fig. S47). In contrast, Karstedt's catalyst could catalyze all these reactions with the yields not correlated with substrate sizes.

With the high sensitivity of **COP1-T-Pt** to steric factors of substrates, we then asked whether **COP1-T-Pt** could selectively catalyze the hydrosilylation on one site among multiple similar functional groups by discriminating the different steric environment between them for high site selectivity[31], which can be achieved by many enzymes but has been regarded as a Holy Grail for artificial catalysts[32–34].

**Site selectivity of COP1-T-Pt.** As shown in Fig. 4, **COP1-T-Pt** showed high site selectivity for alkenes with multiple functional groups. For reaction between triallyl isocyanurate and dimethylphenylsilane, the mono-addition product (compound **12**, 88% yield, Supplementary Fig. 48) was formed predominantly. Only around 4% di-addition product was observed since the cage was not able to accommodate compound **12** and one additional large

dimethylphenylsilane simultaneously. Allyl methacrylate reacted with dimethylphenylsilane to generate a complex mixture when catalyzed by Karstedt's catalyst due to the possibility of hydrosilylation at the allyl, carbonyl and vinyl groups. Interestingly, when **COP1-T-Pt** was used, only the allyl addition product (compound **13**, 95% yield, Supplementary Fig. 49) was formed due to the less steric hindrance at its position, and there was non-detectable amount of methacrylate addition product. The high selectivity also extended to hydrosilylation for other multiple alkene substrates. 2-Methyl-1,4-pentadiene and 2-methyl-1,5-hexadiene, reacted similarly with dimethylphenylsilane at the less-crowded positions also with high selectivity (compound **14**, **15** with the yields of 96% and 85% respectively, Supplementary Figs. 50, 51). The comparison with the performance of Karstedt's catalyst under the same conditions indicated the high selectivity for **COP1-T-Pt** was likely to originate from the confining effect by the cage instead of the electronic effect of the substrates by themselves.

**Fig. 3 Size selective Hydrosilylation for COP1-T-Pt.** Hydrosilylation with different substrate sizes with the demonstration of size-selective effect for **COP1-T-Pt**. Yields were determined by GC-MS.

**COP1-T-Pt** also leads to high site-selectivity for hydrosilylation of conjugated dienes with dimethylphenylsilane. Selective 1,2-selective addition for 1,3-butadienes is rare given their tendency of formation of π-allyl complex with transition metals[31]. However, under the catalysis of **COP1-T-Pt**, dimethylphenylsilane reacted with butadiene to generate solely the mono 1,2-addition product (compound **16**, 85% yield, Supplementary Fig. 52) without the detection of any 1,4-addition product. The reaction between 2,3-dimethyl-1,3-butadiene or myrcene and dimethylphenlsilane also furnished only 1,2-addition product (compound **17**, 75% yield, compound **18**, 84% yield, Supplementary Figs. 53, 54). In contrast, Karstedt's catalyst generated products without any noticeable selectivity. 1,2-Diene, also known as allene, is well-known to generate hydrosilylation products with multiple possible isomers[35]. However, 1,1-dimethylallene reacted with dimethylphenylsilane solely at terminal alkene to form linear allyl silanes (compound **19**, 94% yield, Supplementary Fig. 55) with **COP1-T-Pt**. This is the example of formation of linear allyl silanes by group 10 transition metals. The high regio-selectivity was due to the fact that coordination of the terminal alkene encounters less steric hindrance than others. The unusual selectivity reflects the power of **COP1-T-Pt**'s high steric sensitivity to the substrates, which can direct undreamed-of reaction pathways.

The high site selectivity was also observed when both the hydrosilane and alkene contain multiple functional groups. For example, when 1,4-bis(dimethylsilyl)benzene reacted with triallyl isocyanurate, only the single addition product (compound **20**, 91% yield, Supplementary Fig. 56) was observed for **COP1-T-Pt** catalyst. In contrast, Karstedt's catalyst afforded mixtures of products for compounds **17**, **18** and **19** and gelation for compound **20**. As shown in Fig. 4, for reaction between 1,4-bis(dimethylsilyl)benzene with styrene, only single hydrosilylation product (compound **21**, Supplementary Fig. 57) was obtained in 81% yield without trace amount of double addition product even when excessive alkene molecules were present. Similarly compound **22** (89% yield, Supplementary Fig. 58) was obtained when 1,4-bis(dimethylsilyl)benzene was used. No double addition product was detected.

We then proceeded to evaluate its site-selectivity for alkenes with other functional groups present. Selective hydrosilylation at only the alkene sites in the presence aldehyde group is known to be difficult[7,36]. However, **COP1-T-Pt** provided site selectivity as exemplified by the reaction between dimethylphenylsilane and various allylbenzaldehyde. Hydrosilylation only happened at the alkene position with the aldehyde group completely intact (compound **23**: 97% yield, compound **24**: 93% yield, compound **25**: 70% yield, Supplementary Figs. 59–61). We attributed the high selectivity of terminal alkene over aldehyde to the less steric hindrance at the alkene sites since they were slightly further away from the benzene ring than the aldehydes. We also evaluated the selectivity in the presence of epoxide groups by use of vinylcyclohexane oxide whose hydrosilylation product has the commercial application such as UV-releasing coating and textile finishes. Control experiment with Karstedt's catalyst resulted in gelation when conducted in neat, but **COP1-T-Pt** resulted in clean reaction with the formation of anti-Markovnikov product with the epoxide group untouched (compound **26**, 96% yield, Supplementary Fig. 62), again highlighting its strong preference to the unhindered terminal alkene groups.

Trialkoxysilanes are industrially important[37–39], and in fact most commercial coupling reagents rely on the trialkoxy groups to anchor on various inorganic surfaces, and therefore introduction of alkoxysilane group into organic molecules through hydrosilylation is crucial. The low selectivity and incompatibility with many functional groups for Karstedt's catalyst has limited the scope of trialkoxysilane based materials for applications including as coupling reagents. Encouraged by the result of dimethylphenylsilane shown in Fig. 4, we then proceeded to evaluate the selectivity for the commodity chemical of triethoxysilane, as shown in Fig. 5. Although

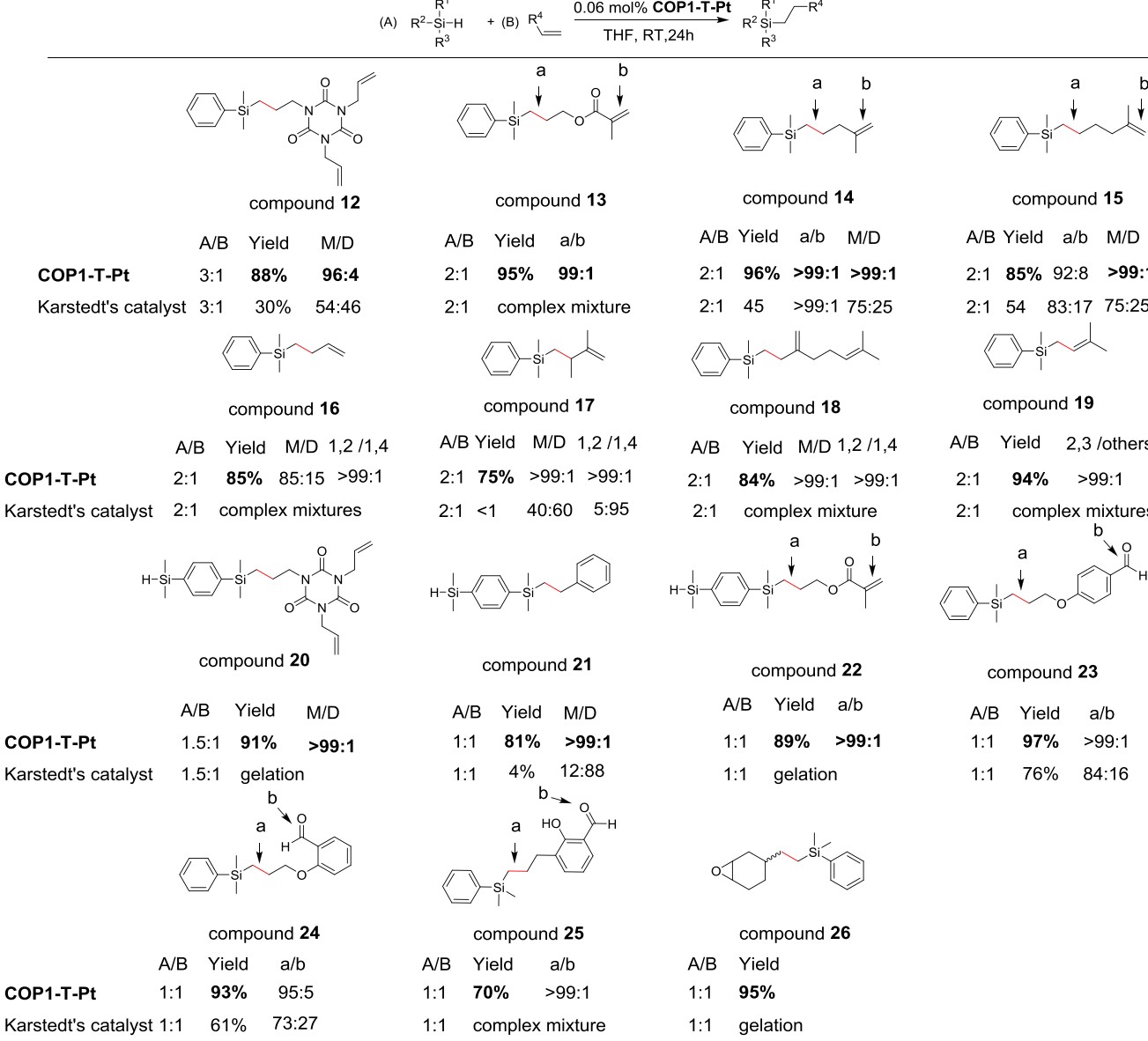

**Fig. 4 Hydrosilylation with multiple functional substrates with the demonstration of site-selective effect for COP1-T-Pt.** Yields (by GC-MS) were based on the moles of the reactants with less initial amounts. The solvent was MeOH for compounds **23**/**24**/**25**. Reaction was in neat for compound **26**.

with a relatively small size, triethoxysialne was able to selectively hydrosilylate allyl methacrylate (compound **27**, 95% yield, Supplementary Fig. 63), 2-methyl-1,4-pentadiene (compound **28**, 96% yield, Supplementary Fig. 64), 2-methyl-1,5-hexanediene (compound **29**, 85% yield, Supplementary Fig. 65), 4-(allyloxy)benzaldehyde, 2-(allyloxy)benzaldehyde and vinylcyclohexane oxide (compound **30**, **31**, **32**; Supplementary Figs. 66–68) with high yields and high selectivity. The use of butadiene as the substrate resulted in the interesting double 1,2 addition due to its small size (compound **33**, 85% yield, Supplementary Fig. 69), which was usually prepared

through the use of uncommon 4-triethoxysilyl-1-butene before our work. Interestingly when conducted in neat, the hydrosilylation reaction between triethoxysilane and acrylic acid also proceeded smoothly although with high catalyst loading (1% mol), resulting in 3-(triethoxysilyl)propanoic acid in one step and high yield (compound **34**, 92% yield, Supplementary Fig. 70). Relative acrylamide derivative (compound **35**, Supplementary Fig. 71) could also be prepared with similarly high yield. The marriage between these commodity materials of large annual production worldwide can provide potential application in wide areas, and this reflected the

**Fig. 5 Hydrosilylation of trialkoxysilanes and substrates with the demonstration of site-selective effect for COP1-T-Pt.** Yields (by GC-MS) were based on the moles of the reactants with fewer amounts. Reactions were in neat for compound **32**, **34**, and **35**.

enabling power of **COP1-T-Pt** unparalleled by any other hydro-silylation catalysts.

## Discussion

Overall, our well-characterized cage-trapped Pt compound exhibited very high catalytical activity for industrially important hydrosilylation reaction. Our mechanism investigation through various techniques including XAS revealed the active specie containing Pt-C and Pt-Si bonds and confirmed the Chalk-Harrod mechanism in our case. This unique catalyst not only showed a broad substrate scope, but also exhibited interesting size-selective catalysis. Furthermore, our catalyst served as a highly site-selective hydrosilylation catalyst for substrates with multiple functional groups, and this is not only scientifically appealing, but also provides access to industrially import alkoxysilanes. We expect that similar Ni or Ir version cage catalyst will provide chances to additional catalysts with high activity and selectivity.

## Methods

**General methods and materials**. Starting materials were purchased from Sigma-Aldrich, Acros, Alfa Aesar, and Fluorochem and used as received without further purification, unless otherwise noted. Dialysis tubes were purchased from Spectrum Laboratories. THF was dehydrated with sodium strap and distilled at 95 °C. All manipulations were carried out in air atmosphere, unless otherwise noted. The transmission electron microscopy (TEM) experiments were recorded on a Hitachi 7650 electron microscope JEM-2100UHR. The EDX elemental mapping and aberration-corrected high-angle annular dark-field scanning TEM (AC-HAADF-STEM) were performed by a JEOL JEM-ARM200F operating at 200 kV. X-ray photoelectron spectroscopy (XPS) was characterized using a Thermo Fisher ESCALAB 250XiPHI Quantera instrument equipped with an Al X-ray excitation source (1486.6 eV). Binding energies were corrected by reference to the C 1 s peak at 284.8 eV. The IR spectra were recorded with KBr pellets on a Bruker EQUINOX

55 FT-spectrometer in the range of $4000-500\ cm^{-1}$, and the $^1$H NMR spectra of most compound were collected in the appropriate deuterated solvents as internal standards on a Bruker Avance Spectrometer with one NMR probe at 300 or 400 MHz. The $^1$H NMR spectra of **COP1-T**, **COP1-T-Pt** and the $^{13}$C NMR spectra, diffusion ordered spectroscopy (DOSY) were characterized on a Varian VNMRS 600 MHz. Chemical shifts are reported in ppm with the solvent resonance as the internal standard (CDCl₃: δ 7.26 ppm, DMSO-D6: δ 2.50 ppm). Data were reported as follows: chemical shift, integration, multiplicity (s = singlet, d = doublet, t = triplet, q = quartet, br = broad, m = multiplet). GC-MS were recorded using Shimadzu GCMS-QP 2010 SE and GC was recorded by Shimadzu GC-2014C. High Resolution Mass spectra (HRMS) were recorded using (1) a mass spectrometer using an Orbitrap XL mass spectrometer (Thermo Fisher Scientific) equipped with an APCI and/or an APPI ionization source or (2) a Finnigan MAT 900 XP double focusing hybrid (EBqQ) mass spectrometer (Bremen, Germany) with a direct insertion probe. Brunauer-Emmett-Teller (BET) surface area and pore size distribution was measured using a Quadrasorb SI-MP instrument; the meso-pore size distribution was calculated from the model of QSDFT. Dynamic laser light scattering (DLS) was performed on the Malvern ZS-90 in THF solution. Elemental analyses (EA) were measured using Flash EA 1112. Single-crystal X-ray diffraction was collected at the beamline BL17B at Shanghai Synchrotron Radiation Facility. The leached platinum content in solution was determined by ICP-OES using an Optima-7000DV after evaporating the organics and dissolving the metal residue in aqua regia. Pt K-edge X-ray absorption spectroscopy (XAS) spectra were collected at the 1W1B-XAFS beam line of the Beijing Synchrotron Radiation Facility and analyzed with the Athena and Artemis software packages. Normalization of the data and background removal were performed in Athena and the X-ray absorption edge energy was calibrated using the spectrum of Pt0 foil. Artemis and its integrated FEFF6 software were used for the extended X-ray absorption fine structure (EXAFS) analysis and the fits were made in R-space with the k²-weighted Fourier-transformed EXAFS data.

**Molecular dynamics simulations**. All-atom molecular dynamics simulations were performed in GROMACS 4.5.5[40]. Molecular topologies and optimized geometries of the reactants were generated using Automated Topology Builder (ATB) (http://atb.uq.edu.au)[41]. Bonded and nonbonded interaction parameters of the molecular topology were modeled using the GROMOS 54A7 force field[42], and particle charges were calculated using the MOPAC method[43]. Lennard-Jones potential was

adopted for Pt atoms, $\varepsilon = 19.41$ kJ/mol, $\sigma = 0.241$ nm[44]. Combination rule was used to obtain the non-bonded interaction parameters between Pt atom and other atoms. GROMOS-96 bond potential was adopted for the coordinate bond between Pt atom and the corresponding carbon atoms in the double bond, $b_0 = 0.22$ nm, $k_b = 10,000$ kJ/mol/nm$^{-4}$. The umbrella pulling was conducted with a harmonic force constant of 100,000 kJ/mol/nm$^{-2}$. We found that reaction path through the vertex, where the carboxyl groups locates, was more preferable. For computation efficiency, all benzene rings in the middle of the eight faces of the cage were constrained to its initial position. The reactant was pulled from the start point to the center of the cage with speed of 0.001 nm/ps. The whole pulling path is 2.5 nm.

During the simulation, periodic boundary conditions were applied in all three dimensions, and temperature was maintained at 300 K using a Nosé-Hoover thermostat. The timestep was 0.0002 ps. Pulling force was extracted every 100 steps for the calculation of the potential of mean force using the Jarzynski's equality[45].

$$\exp[-\beta \Delta G(r)] = \lim_{N \to \infty} \overline{\exp[-\beta w_i(r)]}_N \qquad (1)$$

$$w_i(r) \approx \int_0^r F_i(r') dr' \qquad (2)$$

Here, $i$ indicates the $i$-th pulling simulation, and $r$ indicates the reaction coordinate, and $F$ is the pulling force acted on the reactant, and $w$ indicates the accumulated work done to pull the reactant from the initial point to the current position, and the angle brackets mean average over $N$ trajectories, and $\beta$ is the inverse temperature ($1/k_B T$), and $\Delta G$ is the PMF along the reaction coordinate. Thirty pulling trajectories were used for each PMF calculation.

**X-ray absorption near-edge (XANES) calculation details**. The Pt L3-edge XANES calculations were implemented by FDMNES code in the framework of the multiple scattering theory. This approach used the Green formalism on a muffin-tin potential and the Hedin and Lundqvist approach to get the energy-dependent exchange-correlation potential[46]. In this paper, XANES spectra were calculated for an atomic cluster which centered on the absorption atom Pt. In order to obtain a better simulation effect, the relativistic effect and spin-orbit coupling were considered for calculation. The calculated initial spectra were broadened within the arctangent-dependent width model to account for the energy-dependent broadening that depended on the core level widths and the final state widths.

**Synthesis MOP1. L1** (1.6 g, 1.32 mmol) and Cu(NO$_3$)$_2$·3H$_2$O (4 g, 16 mmol) were dissolved in 240 mL DMF with 4 mL 50% HBF$_4$ aqueous solution and 160 mL CH$_3$CH$_2$OH in a 1000 mL flask. The flask was tightly capped and placed in a 75 °C oven for 36 h to yield turquoise block crystals. MALDI-TOF-MS m/z calcd. for C$_{744}$H$_{792}$Cu$_{12}$O$_{48}$Si$_{24}$Na [M + Na]$^+$: 11962.02, found: 11957.56. Element analysis (EA): C, 73.85; H, 6.65 (calcd.: C, 74.85; H, 6.69).

**Synthesis of COP1-T**. In a sealed tube flask, **MOP1** (1.2 g, 0.1 mmol) and 2nd-Generation Grubbs catalyst (60 mg, 0.72 mmol, 6 eq) in degassed dry THF (6 L) were placed. The mixture was heated at 45°C for 96 h under an argon atmosphere to obtain **MOP1-T**. The solvent was evaporated to about 300 mL. To this solution 300 ml of Na$_2$H$_2$EDTA aqueous solution (0.3 M) was added and then stirred for 12 h at room temperature. The solvent was evaporated to about 300 ml to induce precipitation, which was then washed with water (100 mL × 3) and brine (100 mL × 3). The crude product was recrystallized from THF/n-hexane to obtain **COP1-T** as gray powder 1.05 g (92.6%). $^1$H NMR (600 MHz, DMSO-d6) δ 0.08 (s, 18H), 0.31 (br, 6H), 1.80-0.82(m, 30H), 5.15-4.97 (m, 3H), 7.44-6.95 (m, 30H), 7.87-7.76 (br, 6H), 12.87 (br, 3H), $^{13}$C NMR (150 MHz, DMSO-d6) δ 167.55, 141.08, 140.68, 140.38, 140.24, 140.11, 136.44, 135.78, 132.21, 131.84, 130.93, 130.30, 129.23, 128.40, 127.52, 125.25, 32.47, 28.79, 26.80, 26.36, 23.33, 15.74, −2.87. MALDI-TOF-MS m/z calcd. for C$_{720}$H$_{768}$O$_{48}$Si$_{24}$Na [M + Na]$^+$: 10887.11, found:10887.4. Element analysis (EA): C, 78.78; H, 7.03 (calcd.: C, 79.60; H, 7.13).

**Synthesis of COP1-T-Pt**. To **COP1-T** (7 mg, $6.4 \times 10^{-4}$ mmol) in THF (3 ml) was added the solution of Na$_2$PtCl$_4$·2.5H$_2$O (4 eq, 1.0 mg, $2.6 \times 10^{-3}$ mmol) in THF (1 mL) dropwise in 60 min. During the addition, the colorless solution slowly turned into light yellow. After the addition, the mixture was stirred for 4 h at room temperature. Then dimethylphenylsilane (25 mg, 100 eq) was added, the mixture was stirred for 6 h to afford **COP1-T-Pt** as a dark green solution **S1** ($1.5 \times 10^{-4}$ M, $6 \times 10^{-4}$ M in terms of Pt). Expected Pt content was 0.117 mg/ml and the experimental value by ICP-OES was 0.120 mg/ml. MALDI-TOF-MS m/z calcd. for C$_{720}$H$_{770}$Pt$_4$O$_{48}$Si$_{24}$ [M + H]$^+$: 11645.5 found:11646.9. Element analysis (EA): C, 69.39; H, 6.53. (calcd. **COP1-T-Pt** with 5% H$_2$O: C, 69.58; H, 6.55). **COP1-T-Pt** is slightly hydroscopic in the air, and the EA sample contained 5% water by weight. In fact, TGA analysis of **COP1-T-Pt** also showed the presence of around 4.3% H$_2$O inside as shown in Supplementary Fig. 27.

**Reaction procedure for kinetic study in Fig. 2B**. To the mixture of 1-hexene (8.4 g, 100 mmol) and 2.5 ml **S1** solution of the catalyst prepared as above (diluted by THF down to $1 \times 10^{-5}$ M in terms of Pt atoms) was added different amounts of triethox-ysilane (3.7 mM, 5.6 mM, 9.1 mM, 13.3 mM, 17.2 mM, 21.8 mM, 26.4 mM, 35.7 mM, 70.8 mM, and 138.7 mM respectively), and then the reaction mixture was stirred at

room temperature under argon. The hydrosilylation yields were detected by gas chromatography. The initial reaction rates were calculated when the conversion of 1-hexene was less than 2%. The ICP-OES measurement was performed on the reaction mixture for compound **1** before and after the reaction with the respective values of $0.331 \times 10^{-3}$ mg/ml and $0.329 \times 10^{-3}$ mg/ml. The theoretical value was $0.328 \times 10^{-3}$ mg/ml, and therefore negligible amount of Pt black was formed during our reaction.

**The blocking experiment procedure of hydrosilylation by COP1-T-Pt**. To 1 ml of solution **S1** (the molar amount of carboxylic acid groups was $3.6 \times 10^{-3}$ mmol) was added 100 μl of N,N,N-trimethyl-1-adamantammonium hydroxide in CH$_3$OH (18 mM, $1.8 \times 10^{-3}$ mmol). The mixture was then stirred overnight at room temperature. Then the solution was diluted by THF down to $1 \times 10^{-4}$ M in terms of Pt atoms, and the diluted solution was designated as **S2**. To the mixture of triethoxysilane (5 g, 30 mmol) and 1-hexene (2.6 g, 30 mmol) was added **S2** solution (1 ml, 1 $\times 10^{-4}$ mmol), the mixture with total volume of about 10 ml was then heated at 50 °C under argon. The hydrosilylation yield vs time was detected by GC-MS. Control reaction was performed without the addition of 100 μl of CH$_3$OH solution without N,N,N-trimethyl-1-adamantammonium hydroxide during the preparation of **S2**.

**XAS analysis of COP1-T-Pt catalyst prior to, during and after reaction**. A total of 4 ml of **S1** solution ($1.5 \times 10^{-4}$ M, $6 \times 10^{-4}$ M in terms of Pt) was concentrated down to 1 ml and then added 9 ml xylene, and the obtained solution (**COP1-T-Pt** prior to reaction) was used for XAS measurement at room temperature. THF gradually evaporates during data collection, and the solvent with high boiling points such as xylene has to be used for measurement. To the mixture of 10 mmol triethoxysilane and 10 mmol 1-hexene, 1 ml **S1** solution was added, and the obtained reaction mixture was frozen in liquid nitrogen immediately for XAS analysis (**COP1-T-Pt** during reaction). The above sample was warmed to room temperature for complete reaction, and then the reaction mixture was used for XAS analysis at room temperature (**COP1-T-Pt** after reaction).

**Hydrosilylation reaction for alkene molecules**. Catalyst solution **S1** (1 ml, 0.06 mol% in terms of Pt atoms) was added to a vial containing alkene, silane, and internal standard of C$_{12}$H$_{26}$ (1 mmol each) molecules in argon atmosphere. Experimental details including ratios were provided in scheme 1, 2, and 3. The reactions were monitored by GC-MS and products could be purified by silica gel column chromatography or distillation. For compounds **16, 27, 28, 29, 30, 31, 32** the transformation was so clean that the resulting reaction mixture after vacuum removal of THF solvents could be directly used for $^1$H-NMR spectrum collection. After reaction 10 mg of zinc powder was added for compounds **34** and **35**, and then purified by C-18 silica gel column chromatography. All compounds were characterized with GC-MS to confirm the molecular weight and $^1$H-NMR to confirm the structures. All NMR spectra for the hydrosilylation products (compounds **2–5** in scheme 1, **12–26** in scheme 2, **27–35** in scheme 3) were attached below in this supporting file.

## Data availability

Crystallographic data for **MOP1** (cif and checkcif files) was included in the Supplementary Information files. The cif file was also available in the Cambridge Crystallographic Data Centre with the deposition number of 1950597 [https://doi.org/10.5517/ccdc.csd.cc1lt5m6], and can be obtained free of charge at the CCDC website https://www.ccdc.cam.ac.uk/structures/. Source data are provided with this paper.

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

## Acknowledgements

This work was supported by Youth 1000 Talent Fund KZ37029501 and the 111 Project (B14009). We thank the staffs from BL17B at Shanghai Synchrotron Radiation Facility (SSRF), the biological macromolecule station 3W1A at the Beijing Synchrotron Radiation Facility (BSRF) for help with crystal structure determination and the 1W1B in BSRF, the BL14W1 in SSRF for synchrotron X-ray absorption spectroscopy characterization. This research was supported by the high performance computing (HPC) resources at Beihang University. C.H. acknowledges the support of the National Science Foundation (NSF) Chemistry Research Instrumentation and Facilities Program (CHE-0840277) and Materials Research Science and Engineering Center (MRSEC) Program (DMR-1420073).

## Author contributions

Y.Z.L. conceived the project, designed the experiments and supervised the work. G.H.P. carried out most of the experiments, collected and interpreted the data. C.H.H. solved and refined the X-ray single-crystal structures. S.H. collected the HRTEM data. H.P.L. and Y.J. performed the Gromacs simulation. D.D.Y., C.Q.C., and Q.S.L. contributed to the synthesis of the ligand. N.J.L. performed the XAS simulation with *FDMNES*. L.R.Z. contributed to the interpretation of XAS analysis. L.J. provided comments on the manuscript. Y.Z.L. and G.H.P. cowrote the manuscript.

## Competing interests

The authors declare no competing interest.
