## [Peer Review File · Nature Communications]

REVIEWER COMMENTS

Reviewer #1 (Remarks to the Author):

The paper by Liu and co-workers report the preparation of a nanocage using a strategy previously reported by the Shionoya group (ref 9). Instead of the pyridyl, they report the carboxylate analogue of the building block, which is forming cages after self-assembly using copper paddle wheels. In analogy with the Shionoya group the metal centers are removed after the connection of the building blocks via metathesis. The alkene groups that are part of the cage are used to coordinate to platinum and this is used as catalyst. The materials have been characterized by a variety of techniques, and the novel caged Pt complexes have been used in catalysis. The XANES and EXAFS data look nice, but I cannot judge the quality of it. The properties reported here are very interesting and worth publishing after the following issues have been addressed.

1) The structure of the cage compound is only visualized schematically. An x-ray structure has been reported, but the structure shown in SI is an overview. As the structure of the cage is very important in achieving the typical catalyst properties reported here, the structure should be clearly visualized and discussed. From the schematic picture you get the impression that the Pt atoms are rather at the outside, and it is not clear how to expect cage effects from this structure.

2) They achieve really high activity (78000/h), but at the same time they get substrate selectivity by small changes in the substrate size. For example, the difference between 4 and 5 is very small, yet they give different yields/rate. One would expect that with this difference, the cage pores are relatively small, and that rates are much lower. This should be discussed/explained.

3) They report an intensive list of substrates to demonstrate site selectivity. Whereas for some substrates it is clearly a difference in size only, in other substrates there is also a difference in reactivity. (for example 2713-15)-29, In the later case, selectivity could also be obtained by making use of the reactivity difference. This is not discussed clearly enough, and the authors should be clear on this. In some examples the selectivity is unique, whereas in other cases it is/might not.

4) They should discuss the Michaelis-Menten kinetics in more detail. They suggest that this reflects binding before catalysis, but this can also just be the property of the catalyst. (many metal complexes have substrate coordination before rate determining step and they show MM kinetics.). Also, does the MM constant reflect the binding in the cage. ? how?

5) Related to this, they suggest that the Pt(II)H SiR₃ is the resting state as they observe during the reaction (fig 2F). How is this resting state coupled to the RDS following MM kinetics? This is inconsistent and unclear.

6) I think it would be good to end with a conclusion

7) Judging from the references this is the first example of a catalyst in a cage, which is obviously not the case. It is very disappointing to see that none of the previous work in the area of cage catalysis has been cited. None! Somebody should teach them to do proper literature search and teach them to cite others! Reviews by Mirkin, Raymond/Bergman, Ballester/van Leeuwen, Nolte, Fujita, Reek, Ott on this topic exist and should be cited. Also some of the important original work should be cited. Another example of bad citing: line 153-157 "site selectivityregarded as the holy grail"...with no citations to papers in which this has been achieved by catalyst encapsulation. Ott even wrote a review on this!!

Reviewer #2 (Remarks to the Author):

I enjoyed reading the manuscript 'Biomimetic Caged Platinum Catalyst for Hydrosilylation Reaction with High Site Selectivity' by Liu et al. a lot: Despite the plethora of scientific data it is pleasantly easy to read due to its straight-forward arrangement of ideas and concepts. The reactivity and selectivity of the novel catalyst in hydrosilylation reactions are impressive. I would definitely like to

recommend publication.

There are nonetheless two points I would like to suggest for minor revision:

- (i) The amount in which the catalyst was synthesized should be mentioned in the main text.
- (ii) If commercialized, the catalyst is likely to be placed in the premium range of pricings. The authors are encouraged to provide some examples (from, e.g., medicinal chemistry) for which product purity is required to an extent that justifies the high costs of the catalyst.

Reviewer #3 (Remarks to the Author):

The manuscript of Liu et al. deals with biomimetic caged platinum catalyst for hydrosilylation. XAS is a key method to analyse the structures and mechanisms. The topic is surely of interest, but due to some insufficiencies in the XAS analysis I cannot recommend the paper for publication in its present form. This part needs a major revision to support the conclusions. I will explain this in detail below.

Figure 2: The XANES spectrum of the sample before reaction is assigned to Pt(+I). This is unlikely, as the reference spectrum of H_2PtCl_6 , i.e. a Pt(IV) is given. The decrease in white line intensity is not even a factor of two in the spectrum before reaction, so from this metric a Pt(II) is more reasonable. A XANES deconvolution would strengthen the manuscript here, since it would allow an unequivocal comparison of the WL intensities. During the reaction a +II oxidation state is expected from the proposed structures, but the white line intensity is nearly the same as Pt(IV).

The FT EXAFS spectra must be shown till $R=0$. In general, it appears to me that the background subtraction was not carried out with the necessary care, as some low-R components seem to remain. To much of my regret, I doubt the EXAFS analysis. Despite the fact that a single Pt-H contribution was never fitted before to any EXAFS data because of the minimal scattering amplitude, the spectral changes are very unlikely to reflect the deduced structural changes:

- Going from 2E to 2F: Which signals in the FT correspond to the different contributions? The majority of the signal is formed by the C=C coordination, but none of the FT signals remains at the same distance as in 2E. While this could be an effect of different phase shifts, the reduced amplitude is confusing. 2F was recorded in frozen solution, thus thermal motion should be reduced tremendously. But the amplitude of the FT is reduced which is opposite to the expected behaviour. I disagree that this spectrum proves the claimed mechanism here. The analysis is simply not sound enough to draw this conclusion.

- 2G: Why is the resolution so bad and the amplitude further reduced although now 6 carbon from the C=C are coordinating?

In general the XAS data is not contradicting the conclusions, but it is also not proving them which is mostly due to the sloppy analysis. A revised manuscript should show a table with the fit results including alternative structures that are finally excluded by statistics. I strongly encourage the authors to use highly defined references for their analysis. This makes the conclusions more reliable. Finally I also encourage theoretical XANES calculations to support the assignments.

Dear reviewers:

Thank you for your comments concerning our manuscript entitled “Biomimetic Caged Platinum Catalyst for Hydrosilylation Reaction with High Site Selectivity” (ID: NCOMMS-20-09454). Those comments are all valuable and very helpful for revising and improving our paper, and we have studied them carefully and made corrections which hopefully could meet with the standards for publication. Revised parts have been marked in yellow in the manuscript and supporting information file. The point-to-point replies are as listed as following:

Reviewer #1 (Remarks to the Author):

The paper by Liu and co-workers report the preparation of a nanocage using a strategy previously reported by the Shionoya group (ref 9). Instead of the pyridyl, they report the carboxylate analogue of the building block, which is forming cages after self-assembly using copper paddle wheels. In analogy with the Shionoya group the metal centers are removed after the connection of the building blocks via metathesis. The alkene groups that are part of the cage are used to coordinate to platinum and this is used as catalyst. The materials have been characterized by a variety of techniques, and the novel caged pt complexes have been used in catalysis. The XANES and EXAFS data look nice, but I cannot judge the quality of it. The properties reported here are very interesting and worth publishing after the following issues have been addressed.

- 1) The structure of the cage compound is only visualized schematically. An x-ray structure has been reported, but the structure shown in SI is an overview. As the structure of the cage is very important in achieving the typical catalyst properties reported here, the structure should be clearly visualized and discussed. From the schematic picture you get the impression that the pt atoms are rather at the outside, and it is not clear how to expect cage effects from this structure.

Reply: For a better view of the cage compound and the caged Pt catalyst of

COP1-T-Pt, we provided two videos (video S1 and S2) as part of the supporting information this time for visualizing the inside of the cage. We also provided additional images and discussion in the captions for Figure S7 at page 15 of the supporting information, which are attached below. We hope these additional demonstration could help readers understand the structures of our catalyst.

Figure S7. A) The unit cell shown along the Z axis direction; B, C) Single-crystal structure of MOP1 in space-filling models; D) The schematic drawing of the cage showing an octahedron-shape, which is similar to the one reported by Shionoya group (*J. Am. Chem. Soc.* 131, 11646–11647 (2009)) and Lah group (*Inorg. Chem.* 48, 1281–1283 (2009)); E, F) The corresponding ball and stick model to B and C. The hydrogen atoms are deleted for clarity.

2) They achieve really high activity (78000/h), but at the same time they get substrate selectivity by small changes in the substrate size. For example, the difference between 4 and 5 is very small, yet they give different yields/rate. One would expect that with this difference, the cage pores are relatively small, and that rate are much lower. This should be discussed/explained.

Reply: Thanks for the comment. We agreed with the comment that large substrates should have lower rates than small ones, and this actually was what we observed. The high activity was only claimed for n-hexyltriethoxysilane, not for others.

In fact, the difference between 4 and 5 is surprisingly not as small as one would expect although the tertbutyl group is on the para position, especially considering the catalysis in confined space. The different bulkiness on the para position of benzene ring could have a significant effect on the overall reaction rate. For example, as shown in the following image, Wen group reported that the change from methyl to ethyl group on the para position led to a significant reduction of the yield (from 95% to 78%) (Inorg. Chem. 57, 11157–11164 (2018)).

Reprinted (adapted) with permission from Inorg. Chem. 2018, 57, 17, 11157–11164. Copyright 2018 American Chemical Society.

Entry	Substrates	Products	Yield (%)	TOF (h ⁻¹) ^a
5			95	158
6			78	130

In another paper about catalysis within the confined space in COF (Angew. Chem. Int. Ed. 53, 2878–2882 (2014)), the change was even more significant as shown below. The change from H group in the para position to methyl group led to an overwhelming reduction in the conversion.

Similar size effect was also often observed in enzymatic catalysis. As shown in entry 2,3,4 in the following table, substrate conversion for hydrolase(lipase B)-mediated hydrolysis reaction decreased from 42% to 11% when the substituent groups changed from ethyl to tert-butyl. (Tetrahedron: Asymmetry 22, 47–61(2011)). This showed that the size difference between tert-butyl and ethyl groups was large enough to affect the outcome, and it is fair to expect the difference between tert-butyl and hydrogen group in our case is even bigger.

Table 11
Synthetic scales of hydrolase-mediated hydrolysis of (±)-3a-i

Entry	Ester substrate		Enzyme	ee ^a (%)		Conversion ^b (%)	E value ^b
	R ¹	R ²		Ester	Acid		
1	3a	H	Pseudomonas fluorescens	99 (R)-3a	98 (S)-1a	50	>200
2	3b	H	Candida antarctica lipase B (immob) ^c	65 (S)-3b	90 (R)-1b	42	37
3	3c	H	Candida antarctica lipase B (immob)	26 (R)-3c	98 (S)-1c	21	127
4	3d	H	Candida antarctica lipase B (immob)	12 (R)-3d	99 (S)-1d	11	>200

Reprinted from Tetrahedron: Asymmetry, Volume 22, Issue 1, Deasy et al, Lipase catalysed kinetic resolutions of 3-aryl alkanolic acids, Pages 47-61, Copyright 2011, with permission from Elsevier.

In our case, the para position function group was switched from hydrogen to tertbutyl group, and the yield was change from 92% to 60%, and this seems to be reasonable compared to these examples. We also conducted additional theoretical calculation similar to that shown in Figure S40 to confirm the effect of the tertbutyl group, which was also shown below as Reply Fig. 1. As substrates were pulled into the center of the cage, substrate 5 encountered larger forces than substrate 4.

Reply Figure 1: Theoretical calculation showing the effect of the tertbutyl group when passing through the pore of COP1-T-Pt.

Overall, the opening on the cage doesn't need to be very small to exhibit a difference between these two substrates (4 and 5), and the selectivity for large size molecules and high rate for producing n-hexyltriethoxysilane don't conflict in our opinion. There is a sacrifice of efficiency for substrate with large sizes, and as shown in table 1, 2 and 3 in the main text, the usage of the catalyst was 0.06mol%, compared to the 0.00033 mol% for n-hexyltriethoxysilane.

3) They report an intensive list of substrates to demonstrate site selectivity. Whereas for some substrates it is clearly a difference in size only, in other substrates there is also a difference in reactivity. (for example 13-15, 27-29), In the later case, selectivity could also be obtained by making use of the reactivity difference. This is not discussed clearly enough, and the authors should be clear on this. In some examples the selectivity is unique, whereas in other cases it is/might not.

Reply: Thanks for suggesting the discussion of the reactivity difference, and we provided some clarification here. As shown in table 2 and 3, Karstedt's catalyst failed to provide as good selectivity as our catalyst. Our reasoning is that since there is little steric effect for Karstedt's catalyst, these results should reflect the effect of pure reactivity difference from the substrates for Pt catalyzed hydrosilylation reaction. It seems that pure reactivity difference can't provide such a good selectivity observed in our case. We hope the comparison can help convince you that the major

contributing effect is the cage confining effect instead of reactivity difference.

	compound 12	compound 13	compound 14	compound 15
	A/B Yield M/D	A/B Yield a/b	A/B Yield a/b M/D	A/B Yield a/b M/D
COP1-T-Pt	3:1 88% 96:4	2:1 95% 99:1	2:1 96% >99:1 >99:1	2:1 85% 92:8 >99:1
Karstedt's catalyst	3:1 30% 54:46	2:1 complex mixture	2:1 45% 90:10 75:25	2:1 54% 83:17 75:25
	compound 27	compound 28	compound 29	compound 30
	A/B Yield a/b	A/B Yield a/b M/D	A/B Yield a/b M/D	A/B Yield a/b
COP1-T-Pt	2:1 98% 97:3	2:1 87% 97:3 >99:1	2:1 97% 99:1 >99:1	1:1 96% >99:1
Karstedt's catalyst	2:1 67% 71:29	2:1 53% 68:32 75:25	2:1 47% 70:30 75:25	1:1 74% 82:18

4) They should discuss the Michaelis-Menten kinetics in more detail. They suggest that this reflects binding before catalysis, but this can also just be the property of the catalyst. (many metal complexes have substrate coordination before the rate-determining step and they show MM kinetics). Also, does the MM constant reflect the binding in the cage? how?

Reply: As you suggested, many metal complexes showed MM kinetics due to substrate coordination prior to reaction. However, up to now there has been no hydrosilylation reaction catalyzed by any Pt catalyst to show MM kinetics (Science 298, 204–206 (2002), ACS Catal. 6, 1274–1284 (2016), Catal. Commun. 4, 637–639 (2003), J. Am. Chem. Soc. 121, 3693–3703 (1999)). Therefore, it is likely that the MM kinetics shown in our case is not due to the coordination effect of the substrate to the Pt catalyst. In addition, we also provided the ¹H-DOSY analysis (Figure S27) which indicated the ability of COP1-T to form adducts with alkene and silane molecules, an indication that the cage trapped reactants prior to reaction. The blocking experiment (Figure S28) also supported the catalysis within the cage. With these evidences, we hope to convince that the MM kinetics is consistent with the proposal of the binding in the cage.

5) Related to this, they suggest that the Pt(II)HSiR₃ is the resting state as they observe during the reaction (fig 2F). How is this resting state coupled to the RDS following MM kinetics? This is inconsistent and unclear.

Reply: Thank you very much for the suggestion. We have revised the XAS refinement according to the comments made by reviewer 3. With the new result, the resting state is suggested to be the following compound D shown below (also in Fig. S34), which is also supported by our new XANES simulation work as shown in below (the top middle one in Reply Figure 2 and the reply comments for reviewer 3). This result reflects that the reaction operates through the Chalk-Harrod mechanism with the reductive elimination of alkylsilane from the Pt center as the rate determining step (RDS), and this is also consistent with the previous study (ACS Catal. 6, 1274–1284 (2016), J. Am. Chem. Soc. 124, 9510–9524 (2002), J. Mol. Struct. (Theochem.) 203, 461–462 (1999)). This result is consistent with the MM kinetics since the concentration of the resting state compound is related to the concentration of silane as shown by step II in the mechanism shown below. Higher concentration of silane will lead to higher speed according to the mechanism, and the plateau of reaction speed towards the concentration of silane is expected due to the cage saturation effect, and this is consistent with the MM kinetics.

Figure S34. The Chalk-Harrod reaction mechanism include four steps: (I) Transformation of A into zero-valence Pt species, which probably involves the

reductive silane compounds; (II) Oxidative addition of HSiR_3 ; (III) Insertion of the olefin into the Pt–H bond; (IV) Reductive elimination of alkylsilane. It is believed that step II and III are reversible and that step IV is the rate determining step (RDS).

Reply Figure 2: The comparison between the experimental Pt L3-edge XANES spectra of COP1-T-Pt (prior to, during, after reaction) and the theoretical spectra calculated for the corresponding structures by FDMNES. Spectra calculated for the models proposed in last manuscript were also provided for comparison, confirming the validity of the proposed models in the revised manuscript.

6) I think it would be good to end with a conclusion

Reply: We have ended with a conclusion as below:

Overall, our well-characterized cage-trapped Pt compound exhibited very high catalytical activity for industrially important hydrosilylation reaction. Our mechanism investigation through various techniques including XAS revealed the active specie containing Pt-C and Pt-Si bonds and confirmed the Chalk-Harrod mechanism in our case. This unique catalyst not only showed a broad substrate scope, but also exhibited interesting size-selective catalysis. Furthermore, our catalyst served as a highly site-selective hydrosilylation catalyst for substrates with multiple functional groups, and this is not only scientifically appealing, but also provide unprecedented access to new industrially import alkoxy silanes. We expect that similar Ni or Ir version cage catalyst will provide chances to additional catalysts with high activity and selectivity.

7) Judging from the references this is the first example of a catalyst in a cage, which is obviously not the case. It is very disappointing to see that none of the previous work in the area of cage catalysis has been cited. None! Somebody should teach

them to do proper literature search and teach them to cite others! Reviews by Mirkin, Raymond/Bergman, Ballester/van Leeuwen, Nolte, Fujita, Reek, Ott on this topic exist and should be cited. Also some of the important original work should be cited. Another example of bad citing: line 153-157 “site selectivityregarded as the holy grail”...with no citations to papers in which this has been achieved by catalyst encapsulation. Ott even wrote a review on this!!

Reply: Thanks a lot for the comments, and we have revised accordingly to include these important papers as references 9-22 in the main text to include work by Mirkin, Raymond/Bergman, Ballester/van Leeuwen, Nolte, Fujita, Reek and Otte. The part of main text including these citations was between lines 25 to 28 at page 2 of the main text, which is also provided as following:

many chemists have endeavored to architect caged catalysts⁹⁻¹⁴ from accelerating the reaction rate¹⁵⁻¹⁷, improving selectivity¹⁸⁻²⁰ to alternating reaction pathway^{21,22}, but few of such catalysts has been known to possess both high selectivity and high activity while tolerating a wide range of substrates.

Reviewer #2 (Remarks to the Author):

I enjoyed reading the manuscript 'Biomimetic Caged Platinum Catalyst for Hydrosilylation Reaction with High Site Selectivity' by Liu et al. a lot: Despite the plethora of scientific data it is pleasantly easy to read due to its straight-forward arrangement of ideas and concepts. The reactivity and selectivity of the novel catalyst in hydrosilylation reactions are impressive. I would definitely like to recommend publication.

There are nonetheless two points I would like to suggest for minor revision:

- (i) The amount in which the catalyst was synthesized should be mentioned in the main text.

Reply: Thanks for the suggestion. The amount information has been added at line 8 of

page 4 with the following text:

After copper removal by $\text{Na}_2\text{H}_2\text{EDTA}$, MOP1-T was transformed to COP1-T as white powder (1.05 g, overall 85% yield based on L1, $^1\text{H}/^{13}\text{C}$ -NMR in Fig. S13-14, MALDI-TOF in Fig. S15, EA in S1-7).

- (ii) If commercialized, the catalyst is likely to be placed in the premium range of pricings. The authors are encouraged to provide some examples (from, e.g., medicinal chemistry) for which product purity is required to an extent that justifies the high costs of the catalyst.

Reply: The reviewing paper by Professor Franz (*J. Med. Chem.* **56**, 388–405 (2013)), with the title of ‘Organosilicon Molecules with Medicinal Applications’ provided an exhaustive list of siloxane molecules explored in medicinal chemistry, where purity is crucial. Among the many organosilicon drugs, silanediol-based protease inhibitors such as HIV protease (*Bioorg. Med. Chem. Lett.* **12**, 3625–3627 (2002)), angiotensin-converting enzyme (*Chem. Biol.* **8**, 1161–1166 (2001)), thermolysin (*Med. Chem. Lett.* **14**, 2853–2856 (2004)) and human neutrophil elastase (HNE) (*Expert Opin. Invest. Drugs* **13**, 1149–1157 (2004)) need multi-steps synthesis with very low yields. The synthesis of many drugs involves hydrosilylation reaction (*Met Based Drugs*, **2**, 143–151 (1995), *Acc. Chem. Res.* **46**, 457–470 (2013)). Our caged Pt catalyst may be one suitable candidate to prepare related drug molecules given the high yield and high selectivity.

Reviewer #3 (Remarks to the Author):

The manuscript of Liu et al. deals with biomimetic caged platinum catalyst for hydrosilylation. XAS is a key method to analyse the structures and mechanisms. The topic is surely of interest, but due to some insufficiencies in the XAS analysis I cannot recommend the paper for publication in its present form. This part needs a major revision to support the conclusions. I will explain this in detail below.

1. Figure 2: The XANES spectrum of the sample before reaction is assigned to

Pt(+I). This is unlikely, as the reference spectrum of H_2PtCl_6 , i.e. a Pt(IV) is given. The decrease in white line intensity is not even a factor of two in the spectrum before reaction, so from this metric a Pt(II) is more reasonable. A XANES deconvolution would strengthen the manuscript here, since it would allow an unequivocal comparison of the WL intensities. During the reaction a +II oxidation state is expected from the proposed structures, but the white line intensity is nearly the same as Pt(IV).

Reply: Thanks for the comment, which greatly helped us improve our manuscript. First, we apologize for our carelessness for using the incorrect reference sample and therefore incorrect white line analysis since we found that the reference sample was actually Na_2PtCl_4 (+II for Pt), which we thought was H_2PtCl_6 (+IV for Pt) and therefore the white line intensity analysis went wrong. This time we used PtO_2 and PtCl_4 as the reference samples instead, and the white line intensity of became reasonable for the claim of a low valence state for the sample before reaction (Reply Figure 3). From the analysis of height of white line, it is likely the oxidation state is between 0 and +2. The XPS analysis for the sample before reaction (Figure S29 in the supporting information) also support the oxidation state between 0 and +2. Interestingly, a previous paper published by professor Li (J. Am. Chem. Soc. 140, 7407–7410 (2018)) about the single atom Pt catalyst supported by TiO_2 , also showed similar XPS spectrum and white line analysis, with similar oxidation states.

Figure S29. The oxidation state of Pt in COP1-T-Pt was measured by X-ray photoelectron spectroscopy (XPS). The Pt 4f spectrum was deconvoluted into two peaks at binding energies of 75.8 and 72.4 eV, corresponding to 4f_{5/2} and 4f_{7/2} level, respectively. The peak positions are between those of Pt (II) and Pt (0), indicating Pt atoms carry partially positive charge. (J. Am. Chem. Soc. 140, 7407–7410 (2018)).

Reply Figure 3: Normalized Pt L3-edge XANES spectra of different samples.

In terms of deconvolution, we performed such data process as suggested, and deconvoluted the entire core-hole widths at the L3-edge for samples prior to, during and after reaction, respectively (Reply Figure 4). The Gaussian filter with full width at half-maximum σ value has been chosen by selecting the lowest value that did not give

rise to ringing artifacts in the deconvoluted spectrum.

Reply Figure 4: The comparison between deconvoluted spectra and raw ones.

However, it is necessary to point out that the white line intensities observed in the deconvoluted spectra of above figure are mainly related to the Gaussian filter used in the deconvolution procedure, and no direct physical information can be derived from them (J. Phys. Chem. B 120, 4114–4124 (2016), Phys. Chem. Chem. Phys. 15, 8684–8691 (2013)). An ideal deconvolution with no filter was not possible due to the finite noise level of the experimental data. In fact, as mentioned by Paola (Inorg. Chem. 53, 9778–9784 (2014)), ‘Even if several deconvolution procedures have been developed in the past, deconvoluted XANES signals have been hardly analyzed on a quantitative ground’.

In fact, the non-deconvoluted XANES data could reflect the valence of Pt, and was also consistent with XPS analysis, and therefore we did not use the deconvoluted spectra in the revised version. In addition, it seems to be acceptable by the scientific community that the valence analysis can be derived from the white line analysis from the data without deconvolution as long as no conflict with other analysis (such as XPS) is observed, as seen in many papers published in high impact journals (Nat. Chem. 3, 634–641 (2011), Nat. Commun. 5, 5634 (2014), J. Am. Chem. Soc. 140, 7407–7410 (2018), Metallomics 4, 568–575 (2012)).

2. The FT EXAFS spectra must be shown till $R=0$. In general, it appears to me that the background subtraction was not carried out with the necessary care, as some low- R components seem to remain.

Reply: Thanks for the suggestion. As shown in Figure 2D, the FT EXAFS spectra

have been shown till $R=0$. Background subtraction has been performed to remove the low- R components, as shown by the comparison between the ones before and after background subtraction.

Reply figure 4: Comparison of Fourier transformed (FT) k^2 -weighted $\chi(k)$ -function of the EXAFS spectra for Pt L3-edge spectra before and after background subtraction.

The amount of low R component after background subtraction seems to be comparable to the spectrum reported in the literature, for example, Figure. 2g shown below in one of the papers from Li's group (J. Am. Chem. Soc. 141, 4505–4509 (2019)).

Reprinted (adapted) with permission from J. Am. Chem. Soc. 2019, 141, 11, 4505–4509. Copyright 2019 American Chemical Society.

3. Too much of my regret, I doubt the EXAFS analysis. Despite the fact that a single Pt-H contribution was never fitted before to any EXAFS data because of the minimal scattering amplitude, the spectral changes are very unlikely to reflect the deduced structural changes:

Reply: We appreciate the comment of the Pt-H fitting, and we have revised the analysis accordingly with the help from FDMNES software (Reply Figure 2). The

updated results are shown in Figure 2 (provided below). In fact, the best fitting result suggested that it is highly likely that no PtH was involved in these species, which was actually consistent with previous conclusion that the active species likely involve no PtH bonds (J. Am. Chem. Soc. 124, 9510–9524 (2002)). Thanks again for helping us deepen the understanding of our XAS data.

Reply figure 2: The comparison between the experimental Pt L3-edge XANES spectra of COP1-T-Pt (prior to, during, after reaction) and the theoretical spectra calculated for the models shown within each image. The above three models are used in this manuscript, and the three models below were used in last manuscript. The comparison is meant to show the validity of our new models.

Figure 2. (A) Hydrosilylation reaction between triethoxysilane and 1-hexene catalyzed by COP1-T-Pt. (B) Kinetic study in neat 1-hexene at room temperature, and the concentration of COP1-T-Pt was 1.67×10^{-6} M. The initial reaction rate was measured when the conversion of 1-hexene was less than 2%. The error bar represented the standard error for each initial reaction rate calculated with ten points. (C) Normalized Pt L3-edge XANES spectra of different samples. (D) Fourier transformed (FT) k^2 -weighted $\chi(k)$ -function of the EXAFS spectra for Pt L3-edge spectra. (E, F, G) Quantitative EXAFS fitting curves at R space for COP1-T-Pt prior to reaction, during reaction and after the reaction with the models shown in each panel. (H, I, J) Comparison between the experimental Pt L3-edge XANES spectra of COP1-T-Pt (prior to, during, after reaction) and the theoretical spectra calculated for the corresponding structures. Since XAS only probed the immediate environment around the Pt centers, propene was used instead in for succinctness.

4. - Going from 2E to 2F: Which signals in the FT correspond to the different contributions? The majority of the signal is formed by the C=C coordination, but none of the FT signals remains at the same distance as in 2E. While this could be an effect of different phase shifts, the reduced amplitude is confusing. 2F was recorded in frozen solution, thus thermal motion should be reduced tremendously.

But the amplitude of the FT is reduced which is opposite to the expected behavior.

Reply: As shown in Figure 2D, signals at 1.74 Å, 1.60 Å, 1.69 Å correspond to the

contribution of Pt-C bonds, signals at 1.90 Å COP1-T-Pt prior to reaction correspond to the contribution of Pt-Cl bonds and 2.07Å of COP1-T-Pt during reaction correspond to the contribution of Pt-Si bonds. We appreciate the concern of the distance difference. The positions of the FT signals were also related to the phase shifts, in addition to the nature of the bonds. Other factors including C3 (third-order cumulation), EI (energy shift broadening) and $\Delta E0$ (energy zero shift) can also play a role (J. Synchrotron Rad. 8, 322–324 (2001), J. Synchrotron Rad. 12, 537–541 (2005)). Given the good fit in the XAS analysis and the positive support from FDMNES simulation, the position shift is unlikely to be due to improper models.

About the amplitude, it seems like the temperature doesn't necessarily has a predictable effect on the amplitude. For example, Rh K-edge EXAFS spectra recorded at room temperature and 20 K showed only a small difference in terms of amplitude of the first shell coordination (Figure 6A, J. Phys. Chem. C 123, 14556–14563(2019)). As shown below, for the peak at around 1.7Å, there is some amplitude increase as the temperature decreases (blue line vs yellow line), but it is not significant.

Reprinted
(adapted) with
permission from J.
Phys. Chem. C
2019, 123, 23,
14556–14563.
Copyright 2019
American Chemical
Society

Moreover, in another paper (J. Phys.: Conf. Ser. 430, 012038 (2013)), the amplitude changed randomly for the Pt L3-edge spectrum when the temperature decreased from 600°C to room temperature with the intensity trend of RT < 130 °C < 400 °C < 230 °C, as shown in its Figure 2a, b of its main text (also provided below).

In above example, the amplitude change may be complicated with the local environment change. Temperature doesn't necessarily lead to the height increase, and the overall change can be complicated to an extent that it shows the opposite behavior, and this is probably what happened in our case. This phenomenon deserves further investigation as a separate topic, but at current stage, it seems that this unknown behavior doesn't vitiate the validation of our conclusion for this manuscript.

5. I disagree that this spectrum proves the claimed mechanism here. The analysis is simply not sound enough to draw this conclusion. - 2G: Why is the resolution so bad and the amplitude further reduced although now 6 carbon form the C=C are coordinating?

Reply: As suggested by references, the intensity of the peak is not only affected by the coordination number, but also by amplitude attenuation factor, the fourth-order cumulant and disorder factor (J. Synchrotron Rad. 8, 322–324 (2001), J. Synchrotron Rad. 12, 537–541 (2005)). In addition, the tension around the Pt atoms may also play a role. In fact, for the sample after reaction, the Pt atom was coordinated by three alkene groups. Due to the large size of the hexaphenylbenzene panel, the three alkenes are so spatially separated that they probably cause some tension for them to coordinate the same Pt atom. It is possible that the Pt atom fluctuates between several conformations, or the Pt atom shuttles between different alkene groups within the cage. It is reasonable to postulate that the environment around the Pt centers is somewhat dynamic, and this can lead to complexity of the Pt environment. In fact, the

disorder factor σ^2 was relatively larger for the sample after the reaction than the others, as shown in the fitting table S2, S3 and S4 in the supporting information file.

As known, if the environment around the metal centers becomes complicated, the shape of the peak in EXAFS spectrum usually broadens and the overall intensity decreases, and this seems to be not uncommon. For example, heating will increase the mobility of the peripheral atoms and thus the complexity of environment around the metal centers, and the peak in EXAFS spectrum can be broad. This phenomenon widely exists in XAS measurement, such as, for metals foil measured at high temperature, the corresponding peak will broaden and resolution will deteriorate.

As can be seen from the following images (Figures 5 and 7 of *Geochim. Cosmochim. Acta* 60, 3055–3065 (1996)), the peaks collected at higher temperatures are obviously broader in shape and lower in intensity than the one collected at lower temperatures (the second and fourth panels). Similar results can also be found in other papers such as the one published by Spendelov and Wu (*Adv. Mater.* 30, 1706758 (2018)).

- In general the XAFS data is not contradicting the conclusions, but it is also not proving them which is mostly due to the sloppy analysis. A revised manuscript should show a table with the fit results including alternative structures that are finally excluded by statistics. I strongly encourage the authors to use highly defined references for their analysis. This makes the conclusions more reliable. Finally I also encourage theoretical XANES calculations to support the

assignments.

Reply: We appreciate the suggestion of fitting result table and theoretical XANES calculation. In the revised version, we provided the table for the fitting results as table S2-4 in the supporting information file to support the models used in Figure 2 (Figure S30 to Figure S32, also provided below). In addition to the FDMNES results shown in Figure 2 H-J in the main text, we also provided comparison with other models (Fig. S33, also shown below) to support the validity of proposed models in Figure 2.

Figure S30. The R-factor plot of the corresponding fitted structural models for the COP1-T-Pt catalyst prior to reaction. The green dot represented the best structural model.

Figure S31. The R-factor plot of the corresponding fitted structural models for the COP1-T-Pt catalyst prior to reaction. The green dot represented the best structural model.

Figure S32. The R-factor plot of the corresponding fitted structural models for the COP1-T-Pt catalyst prior to reaction. The green dot represented the best structural model.

Figure S33. The theoretical spectra calculated for the corresponding depicted structures which showed relatively small fitting R-factors during EXAFS fitting but not used in the manuscript. These simulated spectra for these models match less satisfactorily with the best models shown in Figure. 2H-2J. The one highlighted in red boxes are the best one.

Overall, we deeply appreciate your kind comments, and hope our replies provided in this letter can solve your concerns.

Sincerely,

Yuzhou Liu

School of Chemistry, Beihang University

Beijing, 100191, China

E-mail: liuyuzhou@buaa.edu.cn

Tel: 86-10-82316866

2020/08/09

REVIEWER COMMENTS

Reviewer #1 (Remarks to the Author):

The authors put lots of effort in improving the manuscript according to the suggestions of the referees, and in general they addressed most comments.

There is one issue related to the kinetics. They propose a new interpretation of the X-ray data, which leads to the proposed mechanism in S34. The resting state is D (the lowest energy complex before the RDS), and the RDS is proposed to be the reductive elimination. Based on this mechanism one would expect to find a kinetic equation that is zero order in substrate, which means that it doesn't follow Michaelis-Menten kinetics. In line with that, they report in S28 the conversion plot that goes with a straight line to full conversion, (indicating zero order). However, in figure 2 they report a Michaelis-Menten plot that suggests dependency on substrate. This inconsistency should be solved.

It is also strange that the data on which this graph in figure 2 is based is not provided in SI, so we cannot check or reproduce. This should be added. Also, the text should be adjusted accordingly (remove MM kinetics from abstract if it is not)

The second issue that is still not solved is how the size selectivity and activity can be achieved with the same catalyst. They now calculated the energy barrier for entering the cage, which is at least 100 kJ/mol. This is a huge barrier for reactions that are so fast and this seems inconsistent

If these issues are solved the paper can be published

Reviewer #3 (Remarks to the Author):

The authors conducted a considerable amount of work to improve the manuscript and to take my comments about the XAS analysis into account. Where additional analysis was not carried out, my concerns were addressed by elaborate literature examples. Together with the other referee reports on the chemistry, I can now recommend the paper for publication.

Dear reviewers:

Thank you very much for your evaluation and positive feedbacks of our revision entitled "Biomimetic Caged Platinum Catalyst for Hydrosilylation Reaction with High Site Selectivity" (ID: NCOMMS-20-09454). Regarding to the two main concerns, herein we are providing relative information to solve these issues and hope our revised manuscript meet the standards for publication. The point-to-point replies are as listed as the following:

Reviewer #1 (Remarks to the Author):

The authors put lots of effort in improving the manuscript according to the suggestions of the referees, and in general they addressed most comments.

1. There is one issue related to the kinetics. They propose a new interpretation of the X-ray data, which leads to the proposed mechanism in S34. The resting state is D (the lowest energy complex before the RDS), and the RDS is proposed to be the reductive elimination. Based on this mechanism one would expect to find a kinetic equation that is zero order in substrate, which means that it doesn't follow Michaelis-Menten kinetics. In line with that, they report in S28 the conversion plot that goes with a straight line to full conversion, (indicating zero order). However, in figure 2 they report a Michaelis-Menten plot that suggest dependency on substrate. This inconsistency should be solved.

It is also strange that the data on which this graph in figure 2 is based is not provided in SI, so we cannot check or reproduce. This should be added. also, the text should be adjusted accordingly (remove MM kinetics from abstract if it is not)

Reply: We appreciate your comments, and would like to expand our discussion on the kinetics issue regarding to your concerns. In our opinion, it doesn't seem to be right to expect a kinetic behavior with zero-order dependence on the concentration of the silane substrate based on the resting state of being D, as shown in Fig. S34. Based on our mechanism, one could conclude that the concentration of D ([D]) is related to the concentration of HSiR₃ ([HSiR₃]) with the equation as shown in Reply Fig. 1, and therefore overall rate would depend on the [HSiR₃] with the first order approximately, and this is therefore consistent with the Michaelis-Menten plot shown in Figure 2 of the main text.

Reply Figure 1. The different k signs refer to the relative reaction rate constants, and due to the equilibrium between these substrates, such derivations can be obtained as: $[D]=[C]*k_2/k_{-2}$, $[C]=[B]*[\text{HSiR}_3]*k_1/k_{-1}$, and therefore $[D]=[B]*[\text{HSiR}_3]*$

$$k_1 * k_2 / (k_{-1} * k_{-1}).$$

Figure S34 (now S35). The Chalk-Harrod reaction mechanism include four steps: (I) Transformation of A into zero-valence Pt species; (II) Oxidative addition of HSiR₃; (III) Insertion of the olefin into the Pt-H bond; (IV) Reductive elimination of alkylsilane. It is believed that step II and III are reversible and that step IV is the rate determining step (RDS).

The conversion plot shown in S28 (now S30) doesn't contradict with the above conclusion since it was collected with the substrate concentration of about 3000 mM (30 mmol triethoxysilane in 10 ml solution), well above K_m (11.8 mM, for 0.5 V_{max} by definition) measured for Michaelis-Menten plot. Even at 0.9 * V_{max}, the concentration of triethoxysilane was only 106.2 mM. With such a high concentration, the cage was expected to be saturated therefore showing a zero-order kinetics. This is expected for Michaelis-Menten behavior.

The data on which the graph of Figure 2B is based is provided in Fig. S29 at page 29 and the experimental details were provided in section S2-1 at page 8 of supporting information.

2. The second issue that is still not solved is how the size selectivity and activity can be achieved with the same catalyst. They now calculated the energy barrier for entering the cage, which is at least 100 kJ/mol. This is a huge barrier for reactions that are so fast and this seems inconsistent.

If these issues are solved the paper can be published

Reply: We appreciate your comment on the energy barrier calculated with molecular mechanics, and would like to provide feedback.

We understand the importance of activation barrier for reaction kinetics, especially considering the high activity of our catalyst which is more than ten times higher than that of commercial Karstedt's catalyst, and would like to provide

following explanation for your concerns.

The estimation to the energy barriers shown in Fig. S42 C and D was made by the molecular mechanics, on account of the tremendous number of particles involved in the large system we concern currently, instead of the quantum mechanics which considers the detailed interaction on the atomic length scale. The implementation of the quantum mechanics is far beyond our computational capacity, although the calculation precision on the energy is desired. In contrast, the molecular mechanics highly enhances the theoretical predication in the large length scale, due to the application of the available force fields

Steered molecular dynamics has been extensively applied to investigate many biophysical processes, such as the protein-ligand interaction, the transportation of molecules through membrane channels, the unfolding of protein, and other biochemical phenomena. Jarzynski's equality can be used to calculate the free energy along the steered reaction path. However, the quantitative calculation of free energy depends on many factors, e.g., the force constant, the pulling velocity, the choice of the reaction path and the simulation environment settings. [1. J. S. Patel et al. *J. Chem. Inf. Model.* **54**, 470–480 (2014); 2. S. Park et al. *J. Chem. Phys.* **119**, 3559 (2003); 3. H. Xiong et al. *Theor. Chem. Acc.* **116**, 338–346 (2006).]

Only for some special subsets of materials, for example proteins, with highly customized force fields calibrated to a library of abundant experimental data, it may be reliable to use the absolute computed values, otherwise it is advisable to explain the computed values relatively between similar parallel simulations with minor input variations.

In our work, we didn't systematically study the problem partly because of the lack of quantitative experimental data to calibrate the simulation results. We simplified our simulation system by assuming a reaction path from a vertex to the center of the truncated octahedron cage, and using a fixed set of pulling parameters. Without optimization of the parameters, it should be noted that our simulation can only qualitatively characterize the difficulty of molecules entering the cage catalyst.

There are reported examples where the computed values from molecular mechanics were used only for comparison between similar substates, not for comparison with experimental values. Comparison between similar substrates could lead to reasonable conclusions, but the simulated values can be far away from the experimental values in many cases. Two examples were shown below.

Table 1 Collision theory calculations of maximum activation energies to allow complete monolayer formation. Values are shown relative to simulation results. (Visible light corresponds to 160–307 kJ mol⁻¹)

	200 °C, 2 h (kJ mol ⁻¹)	25 °C, 20 h (kJ mol ⁻¹)	Simulated (kJ mol ⁻¹)
Alkene (radical initiation)	140	90	151
Alkene (direct reaction)	110	80	126 250 (ref. 16)
Impurity - oxygen (radical initiation)	100	70	130 (ref. 13)

From page 4863 in *Phys. Chem. Chem. Phys.* 13, 4862–4867(2011).

Table 1 Free energies of solvation in water ($\Delta G_{(g)\rightarrow(aq)}$), *n*-hexane ($\Delta G_{(g)\rightarrow(hxn)}$) and in *n*-hexadecane ($\Delta G_{(g)\rightarrow(hxd)}$) with different charge sets. All values are given in kJ mol^{-1}

Molecule	Charges	$\Delta G_{(g)\rightarrow(aq)}$		$\Delta G_{(g)\rightarrow(hxn)}$		$\Delta G_{(g)\rightarrow(hxd)}$	
		Sim.	Expt.	Sim.	Expt.	Sim.	Expt.
1,3,5-trichlorobenzene	PCM ^a	0.14 ± 0.08	-3.25 (ref. 85)	-28.67 ± 0.10	—	-28.34 ± 0.09	-28.8 (ref. 85)
	Gas ^b	1.19 ± 0.07	—	-28.44 ± 0.08	—	-28.32 ± 0.11	—
Chlorobenzene	PCM ^a	-3.23 ± 0.11	-4.7 (ref. 86)	-18.24 ± 0.04	-21.5 (ref. 86)	-17.61 ± 0.12	-20.9 (ref. 86)
	Gas ^b	-1.91 ± 0.09	—	-18.79 ± 0.04	—	-18.08 ± 0.06	—
Chloroform	PCM ^a	-3.96 ± 0.09	-4.6 (ref. 86)	-13.83 ± 0.06	-15.1 (ref. 87), -13.4 (ref. 86)	-14.04 ± 0.05	-14.1 (ref. 86)
	Gas ^b	1.34 ± 0.08	—	-14.64 ± 0.06	—	-14.91 ± 0.07	—
2,4,5-trichloroaniline	PCM ^a	-26.02 ± 0.18	—	-31.3 ± 0.04	—	-31.90 ± 0.06	—
	Gas ^b	-23.22 ± 0.09	—	-32.67 ± 0.06	—	-32.92 ± 0.07	—
Lidocaine	PCM ^a	-49.19 ± 0.51	—	-46.23 ± 0.07	—	-45.32 ± 0.39	—
	Gas ^b	-30.42 ± 0.44	—	-46.00 ± 0.05	—	-44.65 ± 0.26	—
Methanol	PCM ^a	-19.72 ± 0.06	-21.3 (ref. 88)	-2.16 ± 0.05	-6.2 (ref. 86)	-1.04 ± 0.08	-5.5 (ref. 86)
	Gas ^b	-15.76 ± 0.04	—	-4.31 ± 0.06	—	-3.13 ± 0.05	—

^a Effectively polarized charges obtained from a polarizable continuum model (PCM) (see text for more information). ^b Gas phase charges.

Reprinted from
*Phys. Chem.
Chem. Phys.*,
2013, 15,
4677–4686

From page 4681 in *Phys. Chem. Chem. Phys.* 15, 4677–4686 (2013).

Actually Maiti mentioned in his paper that “this is really a hard problem from the computational point of view. Because of slow kinetics and rich conformational degrees of freedom of the polynucleotides on the CNT surface, there will be a variety of ground-state equilibrium configurations for each case, as has been demonstrated by us and earlier by several groups. Therefore, calculating accurate PMF requires extensive averaging from all of these various equilibrium conformations.” [*ACS Appl. Mater. Interfaces* 9, 35287–35296 (2017)]. Herein PMF refers to potential of mean forces calculation by molecular dynamics.

Secondly, in light of the comment, we measured the experimental activation energy of the hydrosilylation of *n*-hexene with triethoxysilane at different temperatures (40, 45, 50, 55 °C), and obtained the value of about 40 kJ/mol (Reaction procedure in S2-5), which was much lower than that of Karstedt catalyst (~ 60 kJ/mol) [*ACS Catal.* 6, 1274–1284 (2016)] with similar substrates. This low experimental activation energy is consistent with the claimed high activity for our catalyst. Detailed information for this additional experiment is provided as Fig S24 in the supporting information.

Figure S24. Kinetic investigation of the hydrosilylation of n-hexene with triethoxysilane. The experimental detail was provided in S2-5.

We deeply appreciate your kind comments, and hope our replies provided in this letter can solve your concerns.

Sincerely,

Yuzhou Liu
 School of Chemistry, Beihang University
 Beijing, 100191, China
 E-mail: liuyuzhou@buaa.edu.cn
 Tel: 86-10-82316866
 2020/10/13

REVIEWERS' COMMENTS

Reviewer #1 (Remarks to the Author):

The authors provided additional data and comments to my final remarks on kinetics and molecular modelling results.

1) with respect to the kinetics I conclude the following. Compound B is the resting state (lowest energy before the rate determine step) and as a result they find an order in substrate. If the first two steps to form D are reversible (and then reductive elimination as a rds) the reaction follows MM kinetics. Indeed at high concentration, the equilibrium shifts to species D (and under these conditions they have zero order in substrate), which is the species found by in situ spectroscopy as these experiments are done at high concentrations. I think the authors will agree with this, as the only difference in explanation is the fact that D is not the resting state, but B.

2) In their explanation they state that the calculations are by far not accurate enough to determine energy barriers. This should be CLEARLY stated in the SI and or paper.

With these minor changes the paper can be accepted and I congratulate the authors with their nice results.

Dear reviewer 1:

Thank you very much again for your evaluation and positive feedbacks of our reversion entitled "Biomimetic Caged Platinum Catalyst for Hydrosilylation Reaction with High Site Selectivity" (ID: NCOMMS-20-09454). Regarding to the two main concerns, herein we are providing relative information to solve these issues the point-to-point replies are as listed as the following:

The authors provided additional data and comments to my final remarks on kinetics and molecular modelling results.

1) with respect to the kinetics I conclude the following. Compound B is the resting state (lowest energy before the rate determine step) and as a result they find an order in substrate. If the first two steps to form D are reversible (and then reductive elimination as a rds) the reaction follows MM kinetics. Indeed at high concentration, the equilibrium shifts to species D (and under these conditions they have zero order in substrate), which is the species found by in situ spectroscopy as these experiments are done at high concentrations. I think the authors will agree with this, as the only difference in explanation is the fact that D is not the resting state, but B.

Reply: We appreciate your comments, and agree with you on the sense that the resting state of B or D don't alter the first order dependence on the concentration of silane substrate. Thank you very much for putting your valuable thoughts in our manuscript.

2) In their explanation they state that the calculations are by far not accurate enough to determine energy barriers. This should be CLEARLY stated in the SI and or paper.

With these minor changes the paper can be accepted and I congratulate the authors with their nice results.

Reply: Thank you very much for remaining us, and we have added the statement in the legend of Supplementary figure 47 as:

It should be noted that the simulation can only qualitatively characterize the difficulty of molecules entering the cage catalyst.

We deeply appreciate your kind comments, and hope our replies provided in this letter can solve your concerns.

Sincerely,

Yuzhou Liu
School of Chemistry, Beihang University
Beijing, 100191, China
E-mail: liuyuzhou@buaa.edu.cn
Tel: 86-10-82316866
2020/11/1